# Entropy favors heterogeneous structures of networks near the rigidity threshold

Le Yan [ID] [1]

The dynamical properties and mechanical functions of amorphous materials are governed by their microscopic structures, particularly the elasticity of the interaction networks, which is generally complicated by structural heterogeneity. This ubiquitous heterogeneous nature of amorphous materials is intriguingly attributed to a complex role of entropy. Here, we show in disordered networks that the vibrational entropy increases by creating phase-separated structures when the interaction connectivity is close to the onset of network rigidity. The stress energy, which conversely penalizes the heterogeneity, finally dominates a smaller vicinity of the rigidity threshold at the glass transition and creates a homogeneous inter-mediate phase. This picture of structures changing between homogeneous and hetero-geneous phases by varying connectivity provides an interpretation of the transitions observed in chalcogenide glasses.

[1] Kavli Institute for Theoretical Physics, University of California, Santa Barbara, CA 93106, USA. Correspondence and requests for materials should be addressed to L.Y. (email: lyan@kitp.ucsb.edu)

L acking long-range order, amorphous materials are fully governed by their microscopic structures. Increasing evidence indicates that the structural elasticity of such materials correlates with their dynamical properties and mechanical functions, such as the suddenly slowing relaxations of glasses[1–4] and the allosteric regulation of proteins[5, 6]. A crucial factor behind the structural disorder that controls the linear elasticity of a structure is the average number of constraints $n$ of its interaction network and the rigidity transition associated with tuning $n$[7]. At the Maxwell point $n_c = d$[8], which is the minimum number of constraints per particle to avoid floppy modes in spatial dimension $d$, both the elastic moduli and self-stresses vanish, accompanied by a vanishing onset frequency $\omega^*$ of the soft vibrations on the so-called boson peak[9–11]. However, it is questionable whether these results obtained in homogeneous networks apply to heterogeneous network structures, which may be fundamental.

Chalcogenides, for example, are network glasses composed of chemical elements with different covalent valences $r$, proportional to which the number of covalent constraints $n$ varies. Rather than a point threshold $r_c = 2.4$[12, 13], a range of singular features, named the intermediate phase, bridges the well-connected stressed and poorly coordinated floppy phases, as observed in experiments[14–16] and reproduced in molecular dynamics simulations[17–19]. Inside the phase, the non-reversible heat, a glass-transition equivalent of the latent heat, vanishes[14], which is associated with a vanishing stress heterogeneity[15] and a minimal molar volume[16]. All of these measurements are discontinuous when entering the phase from either side[16]. The critical point observed in random networks[20–22] (Fig. 1a), which allow fluctuations in local connectivities, fails to capture the nature of the intermediate phase. Emerging in self-organized networks to reduce the energetic costs of self-stressed states[23, 24] (Fig. 1b), the rigidity window with distinct onsets of rigidity and self-stress promisingly maps to a critical range like the intermediate phase; however, the stronger heterogeneity inside the critical window actually contradicts the experimental observations, and the window is also sensitive to the appearance of prevailing perturbations such as van de Waals forces[25]. In fact, a rather odd feature is the heterogeneous nature away from the threshold, outside of the intermediate phase. What causes the heterogeneity beyond the local fluctuations?

Recent achievements[26, 27] indicate that the entropy, a synonym of 'disorder', leads to order and heterogeneity in many cases, including the gas-crystal phase separation in colloid-polymer mixtures[28, 29] and the open lattice structures of patchy particles[30, 31]. The key components that allow for this comprehensive role are the high degeneracy of configurations and the separation of degrees of freedom carrying entropy from the ones assembling structures. In amorphous networks, configurations are inherently degenerate. Floppy and soft modes on boson peaks store significant amounts of vibrational entropy[32], particularly close to $n_c$; thus, they inevitably shape the network structures.

In this communication, we investigate the role of entropy in regulating network structures and show the appearance of phase-separated heterogeneous structures ruled by a critical point at the rigidity threshold. We then confirm the appearance of a homogeneous intermediate phase when stress energy dominates at low temperature. Finally, we apply the results to chalcogenides and discuss several experimental evidence of phase separation.

## Results

**Network model**. To illustrate our main result of an entropy-induced phase-separated connectivity range near the rigidity threshold, we consider a network model on a two-dimensional triangular lattice with periodic boundaries, nevertheless, the result

and its derivation depend neither on the dimension nor on the lattice structure. On lattice, a particle at each of $N$ nodes can be wired to at most all of its six neighbors, corresponding to the maximal constraint number $n_m = 3$. Following ref.[20], we randomly perturb the locations of lattice nodes to avoid straight lines that lead to non-generic singular modes, as shown in Fig. 2a. The key assumption of the model is the separation of energy scales such that we can consider the network of the stronger interactions such as the covalent bonds in chalcogenides and treat the weaker ones such as van der Waals forces as perturbations. In the simplest construction, a network configuration $\Gamma$ is defined by the allocation of $N_s \equiv nN$ linear springs of identical stiffness $k$ on the $n_m N$ possible links. Different configurations are probed by relocating one random spring (red solid) to an unoccupied (blue dashed) link at a time, as illustrated in Fig. 2a, such that their number is fixed by a given average number of constraints $n$, similar to rearranging atoms of different valences in network glasses. The different configurations are sampled with probabilities proportional to the Boltzmann factor $\exp(-F/T)$ using the Metropolis algorithm, which is documented together with the model parameters in Methods section. Given configuration $\Gamma$, its free energy is

$$F(\Gamma) = H_0(\Gamma) - TS_{\mathrm{vib}}(\Gamma), \qquad (1)$$

where vibrational entropy $S_{\mathrm{vib}}$ quantifies the volume of thermal vibrations near the mechanical equilibrium of $\Gamma$[31–33],

$$S_{\mathrm{vib}}(\Gamma) = -\frac{1}{2}\ln\det\frac{\mathcal{M}(\Gamma)}{T} = -\sum_\omega \ln\omega + c \qquad (2)$$

which depends on $\omega^2$–the eigenvalues of Hessian $\mathcal{M}$ and a $\Gamma$-independent number $c$. $H_0$ is the self-stress energy of $\Gamma$ at equilibrium. We introduce frustrations by imposing that the rest length of the spring $\gamma$ positioned at the link $\langle ij \rangle$, $l_\gamma = r_{\langle i,j \rangle} + \epsilon_\gamma$, differs from $r_{\langle i,j \rangle}$, the spacing between neighboring nodes $i$ and $j$, by a mismatch $\epsilon_\gamma$ assigned from a Gaussian distribution of zero mean and variance $\epsilon^2$. In the small frustration limit, where $\epsilon$ is much smaller than the lattice constant, we compute $\mathcal{M}$ and $H_0$ in the linear approximation, as derived in the Supplementary Note 1 and refs.[4, 25, 34].

We include perturbations of non-specific but weaker interactions by connecting all six second neighbors on the lattice with springs of stiffness $k_w \ll k$. At this high connectivity, they act approximately as isotropic potentials of effective stiffness $\alpha = \frac{6k_w}{dk} \ll 1$ time of $k$. These weak forces hence set a finite vibration volume for floppy modes while leaving the other modes nearly untouched, as illustrated in Fig. 2b, c.

**Entropy favors phase separation**. As shown in Fig. 1c, in the limit of no self-stress penalty $\epsilon = 0$ and thus no energy regulation $H_0 = 0$, entropy-favored networks present a phase separation into two phases, a highly coordinated stressed cluster ($n > n_c$ dark green) and a floppy phase formed by the remaining clusters ($n < n_c$ blue), near $n_c$, distinct from the homogeneous structures in Fig. 1a, b, where the percolating rigid cluster would appear indistinguishable from the remainder if the color code and the pivots are removed in Fig. 1. This phase separation is captured by a long-range correlation of the local constraint number and a bimodal cluster size distribution (a system-size stressed cluster plus small ones in the floppy phase) in contrast to a continuous one[35], as shown in Fig. 3a, b.

Due to the phase separation, the network rigidity arises in a discontinuous fashion as the stressed cluster percolates—growing from an island inside the floppy sea to a continent enclosing floppy lakes. This percolation occurs at a constraint number $n^*$

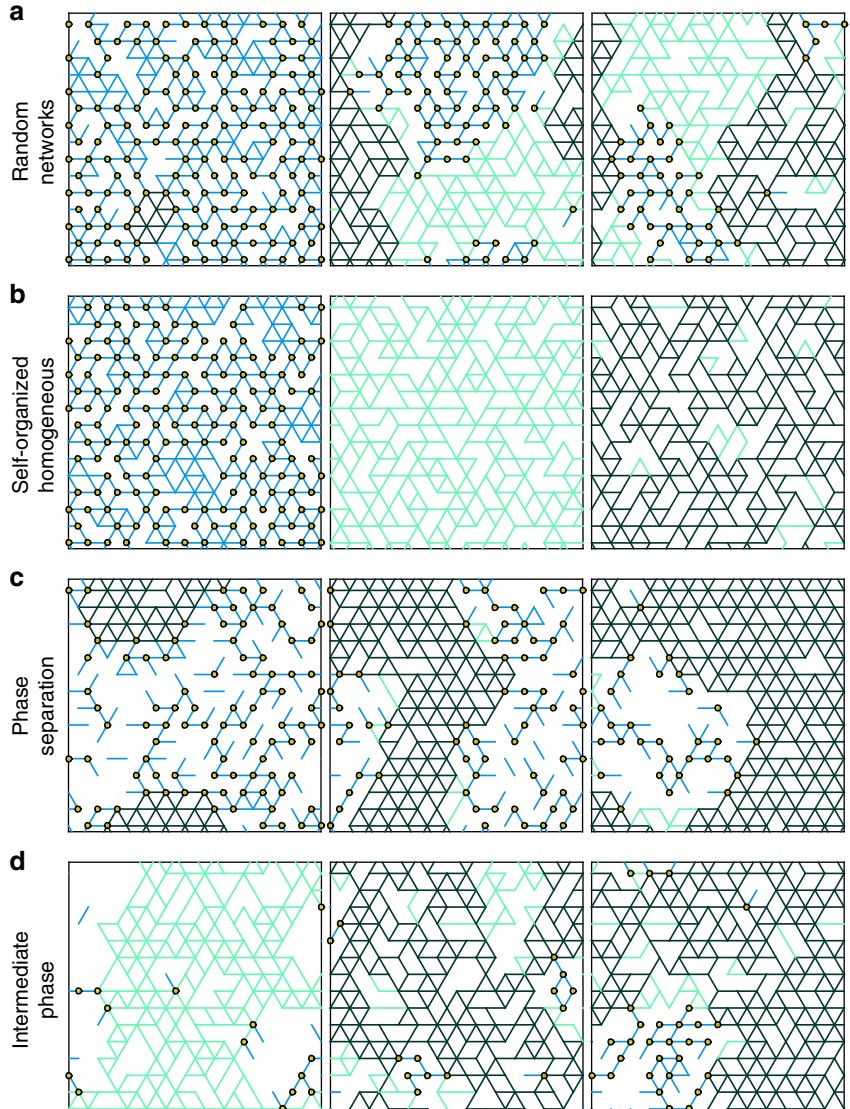

**Fig. 1** Scenarios of rigidity transition in elastic networks. Typical structures for **a** random networks, **b** self-organized networks, **c** entropy-favored networks, and **d** equilibrated networks at $T_g$. From left to right, are structures below, at, and above the rigidity transition threshold. For illustration, we implement the pebble game algorithm[20, 46] to decompose the networks into irreducible rigid clusters, unstressed (blue and green) and stressed (dark green), which are connected by pivots[47], shown as yellow circles. The connections in the percolating cluster are colored in green (light and dark) and the remainder clusters making the floppy regions are in blue

different from $n_c$, which is captured by a discontinuous $P_\infty$, the probability of springs in the percolating cluster, as shown in Fig. 3c. In Fig. 3d, the bulk modulus $K$ shows a trend to jump at $n^*$, whereas the shear modulus $G$ vanishes.

**Phase diagram**. Why does entropy alone favor a floppy-rigid phase separation? As the degrees of freedom carrying vibrational entropy (particles) disconnect from the ones coding the configuration (springs), the total entropy increases by creating floppy modes in the floppy subpart of the network by confining springs in the stressed counterpart, particularly when this spring redistribution costs little configurational entropy near the rigidity threshold. When the self-stress energy is not participating, the balance between the vibrational entropic gain and the configurational cost determines the stability of the separation.

Consider a separation into a homogeneous rigid phase and a floppy phase of volume fractions $V_r$ and $V_f$ controlled by the constraint numbers $n_r$ and $n_f$, as illustrated in Fig. 4a. The configurational entropy is the entropy of mixing springs and

vacancies summed over the two phases,

$$\frac{S_{\text{conf}}}{N} = s_{c,0} + V_r \left( n_r \ln \frac{n_m}{n_r} + (n_m - n_r)\ln \frac{n_m}{n_m - n_r} \right) + V_f \left( n_f \ln \frac{n_m}{n_f} + (n_m - n_f)\ln \frac{n_m}{n_m - n_f} \right), \quad (3)$$

plus $s_{c,0}$, the entropy from the boundary contribution, which vanishes in the thermodynamic limit. As the extra vibrational entropy gains from the floppy modes, let us assume that the vibrational entropy is proportional to the number of floppy modes,

$$\frac{S_{\text{vib}}}{N} = s_{v,0} + V_f(n_c - n_f)\Lambda, \quad (4)$$

changing by $\Lambda$ per floppy mode. As shown in the Supplementary Note 2, this assumption is approximately valid in the model and per mode entropy gains $\lambda = -\frac{1}{2}\ln\alpha + \langle\ln\omega\rangle > 0$, where $\langle\ln\omega\rangle$ is

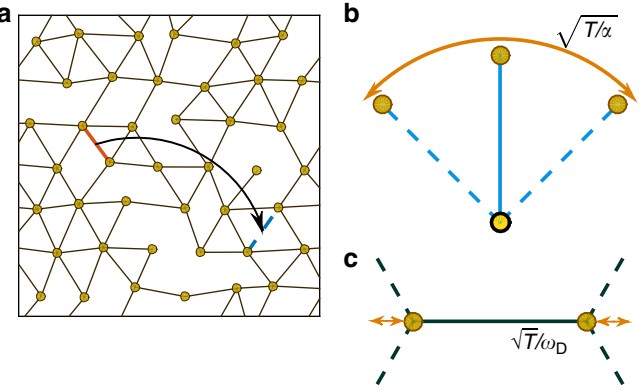

**Fig. 2** Illustration of the network model and vibrational space. **a** The spring connections between perturbed neighbor nodes on triangular lattice, shown as solid lines, define a network configuration. Weak springs (not shown) connect all second neighbors. A new configuration is sampled by moving a randomly selected strong spring in red to a random vacant lattice link shown in blue dashed line. Given a configuration, thermal vibrations correspond to **b** a floppy mode and **c** a Debye-frequency mode at temperature $T$

the spectrum-average entropy of non-floppy modes. Henceforce, we use the convention of the large $\Lambda$ as a parameter in the formalism and the small $\lambda$ as the actual entropic gain in the model.

Constrained on the total volume $V_f + V_r = 1$ and the average constraint number $n_f V_f + n_r V_r = n$, the total entropy $S_{vib} + S_{conf}$ is optimized with

$$\frac{n_r}{n_m} = \frac{e^{-\frac{\Lambda n_c}{n_m}} - 1}{e^{-\Lambda} - 1}; \tag{5}$$

$$\frac{n_f}{n_m} = \frac{1}{1 + e^{\Lambda}\left(\frac{n_m}{n_r} - 1\right)} = \frac{e^{\frac{\Lambda n_c}{n_m}} - 1}{e^{\Lambda} - 1}; \tag{6}$$

$$V_r = \frac{n - n_f}{n_r - n_f}. \tag{7}$$

Since $V_r \in [0, 1]$, the heterogeneous phase exists in the self-consistent range $n \in [n_f, n_r]$, which is very wide $\frac{n_r - n_f}{n_f} \sim \lambda \sim -\frac{1}{2}\ln\alpha$ for practical $\alpha$. The boundaries $n_f(\Lambda)$ and $n_r(\Lambda)$ define the heterogeneous separation phase in the phase diagram in Fig. 4a.

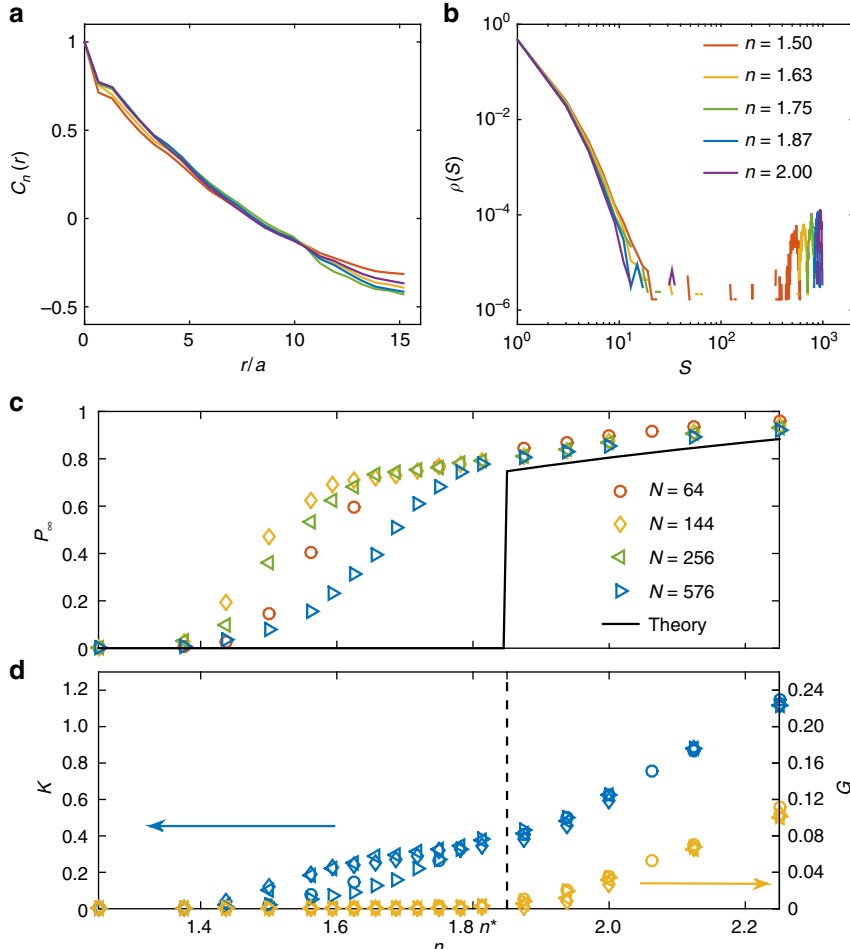

**Fig. 3** Features of network structures optimizing the total entropy. **a** Spatial correlation of connectivity $C_n(r) = \left(\langle n(r)n(0)\rangle - \bar{n}^2\right)/\left(\langle n(0)^2\rangle - \bar{n}^2\right)$ for entropy favored networks, $N = 576$. **b** Probability distribution of rigid cluster sizes $\rho(S)$, collapse for a wide range in $n < n_c$. **c** Probability in the percolating cluster $P_\infty$, and **d** Bulk modulus $K$ and shear modulus $G$ vs. constraint number $n$ for various system sizes $N$, $\alpha = 0.0003$. The black solid line is theoretical prediction for the thermodynamic limit $N \to \infty$. $\lambda \approx 3.3$, so $n_f \approx 0.94$, $n_r \approx 2.76$, and $n^* \approx 1.85$, fitted by Eqs. (5), (6) and (7)

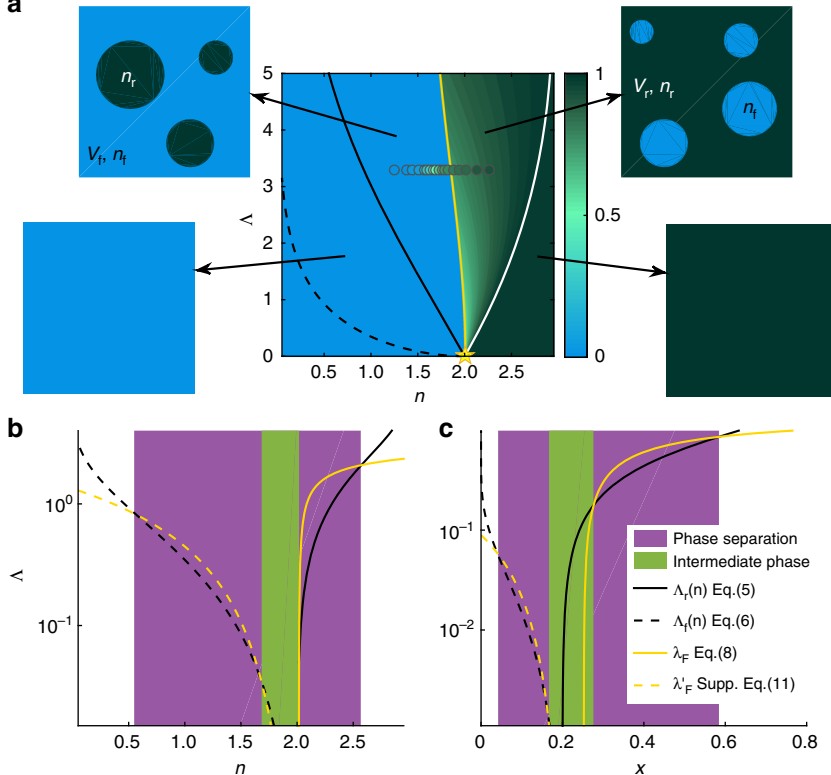

**Fig. 4** Phase diagrams of network structures near the rigidity threshold. **a** Phase diagram of the model in $n$-$\Lambda$ space. The star sign marks the critical point at $n_c = d$, $\Lambda_c = 0$. The phase boundaries shown by black and white solid lines defined in Eqs. (5), (6) and (7) separate the heterogeneous phase mixed of floppy and stressed regions and homogeneous phases as illustrated. The floppy parts are in light blue and the stressed parts are in dark green. The dashed line shows the phase boundary Eq. (10) towards a floppy-isostatic mixture phase. The color bar labels the probability of a bond in the percolating cluster $P_\infty$, which jumps at the yellow line $n^*(\Lambda)$ when the rigid phase reaches half volume fraction. Numerical data of $N = 576$ $\lambda = 3.3$ are shown in circles. **b** Phase diagram of model at $T_g$ with $\Lambda$ in log scale. The black solid and dashed line reproduce the phase boundaries (white solid and black dashed) in **a**. On the rigid side $n > n_c$, when the free energy loss at $T_g$ given by Eq. (11) shown by yellow solid line is above the boundary, the heterogeneous networks appear in equilibrium. When $n < n_c$, the phase separation is stable when the yellow dashed line showing the free energy loss given by Supplementary Eq. (19) goes beyond the heterogeneous boundary. **c** Same phase diagram showing the intermediate phase for compounds $A_xB_{1-x}$. The purple regions show the range of heterogeneous phases, and the green region is the homogeneous intermediate phase

Analogous to the classical spontaneous magnetization and gas–liquid phase separation, the entropy-induced floppy-rigid separation is governed by a critical point at $\Lambda = 0$ and $n = n_c$, where the entropy gain $\Lambda$ plays as the relevant parameter like temperature $T_c - T$ and the average constraint number $n$ is the order parameter akin to the mean magnetization $M$ in ferromagnet or the mean density $\rho$ in gas–liquid separation. Close to the critical point $\Lambda = 0$ and $n = n_c$, the free energy follows,

$$F = -TS_{vib} \sim \Lambda^{2+\alpha}. \quad (8)$$

The counting approximation, Eq. (4), gives $\alpha = -1$. The order parameter scales as,

$$n_{r,f} \sim \Lambda^\beta. \quad (9)$$

The mean-field solution, Eqs. (5), (6) and (7) implies $\beta = 1$. Both exponents are different from the standard Landau theory.

A typical size is thus determined by the critical scaling approaching $(0, n_c)$. In the separation range, the network structure presents the dominant phase $(V > 1/2)$ with droplets of the subdominant one of this characteristic size. Global rigidity arises when the rigid phase becomes dominant at $n^* = (n_f + n_r)/2$, as indicated by the yellow line in Fig. 4a.

**Self-stress penalty and homogeneous intermediate phase.** When creating self-stressed states is prohibited[23, 24], phase separation can still arise for $n < n_c$ due to an entropy gain of additional soft modes on the boson peak in isostatic structures. Per degree of freedom in isostatic volume $V_c$, the vibrational entropy increases $\Lambda' \equiv \frac{\partial S_{vib}/N}{d\partial V_c}$, positive as shown in the Supplementary Note 2. This gain from isostatic structures leads to a separation between an isostatic phase and a floppy phase, as illustrated in Fig. 1j. The corresponding phase boundary follows

$$\Lambda' = \ln\frac{n_c}{n_f} + \left(\frac{n_m}{n_c} - 1\right)\ln\frac{1 - n_c/n_m}{1 - n_f/n_m} \quad (10)$$

shown as the white dashed line in Fig. 4a.

Because reducing the self-stress energy tends to level the connection distribution[25], when the energetic cost $H_0$ competes with the entropic gain, a homogeneous intermediate phase can develop inside the heterogeneous gap at low temperature. In Fig. 1d, we depict the typical network structures equilibrating the total free energy Eq. (1) at the glass transition temperature $T_g$. From left to right, which correspond to below, at, and above $n_c$, the networks are floppy-isostatic heterogeneous, homogeneous, and floppy-stressed heterogeneous, respectively.

At temperature $T$ (in the energy unit $k\epsilon^2 \equiv 1$), each self-stressed state contributes an independent direction to store energy[4, 34]. Noticing the duality between self-stressed states and

floppy modes[36], a free energy loss per floppy mode substitutes the entropy gain $\Lambda$ in Eq. (4),

$$\Lambda \rightarrow \lambda_{\mathrm{F}}(T) = \lambda - \frac{1}{2}\ln\left(1 + \frac{1}{T}\right) \qquad (11)$$

(see the Supplementary Note 3 for the derivation). The self-consistent condition of floppy-rigid phase separation breaks down when $\lambda_{\mathrm{F}}(T) \leq \Lambda(n)$, the phase boundary in Eq. (5). Relying on the insights of the elastic models[37], we apply a glass transition temperature that is proportional to the shear modulus, $T_{\mathrm{g}} \propto G$, whose analytical form is derived in the Supplementary Note 4. When $n > n_{\mathrm{c}}$, $T_{\mathrm{g}} \sim n - n_{\mathrm{c}}$[4, 34], $\lambda_{\mathrm{F}}$, shown as the blue solid line in Fig. 4b, reenters the homogeneous phase when $n$ decreases close to $n_{\mathrm{c}}$, $n_{\mathrm{r}} - n_{\mathrm{c}} \sim \alpha$, defining the threshold of the homogeneous intermediate phase on the rigid side.

When $n < n_{\mathrm{c}}$, $T_{\mathrm{g}} \sim \alpha \ll 1$[4, 34], the self-stress prohibited situation applies. Derived from a flat mode density approximation[4, 38] in the Supplementary Note 3, the free energy loss per isostatic volume, shown as the blue dashed line in Fig. 4b, surpasses the heterogeneous boundary Eq. (10) in the dashed line at $n_{\mathrm{c}} - n_{\mathrm{f}} \gtrsim \sqrt{\alpha}$, giving the transition from the intermediate phase on the floppy side. Altogether, as the connectivity increases, the network structures change from homogeneous floppy to heterogeneous floppy-isostatic to intermediate homogeneous marginal to heterogeneous floppy-stressed and finally to homogeneous stressed, as depicted in Fig. 4b.

**Relative entropy.** This floppy-rigid phase separation has a general information theory implication. Rewriting the phase boundaries $n_{\mathrm{f}}(\Lambda)$ and $n_{\mathrm{r}}(\Lambda)$ in Eqs. (5), (6) and (7) in terms of relative entropies[39], $D(p|q) = p\ln\frac{p}{q} + (1-p)\ln\frac{1-p}{1-q}$, we find that

$$D\left(\frac{n_{\mathrm{c}}}{n_{\mathrm{m}}}\bigg|\frac{n_{\mathrm{f}}}{n_{\mathrm{m}}}\right) = D\left(\frac{n_{\mathrm{c}}}{n_{\mathrm{m}}}\bigg|\frac{n_{\mathrm{r}}}{n_{\mathrm{m}}}\right); \qquad (12)$$

$$(n_{\mathrm{c}} - n_{\mathrm{f}})\Lambda = n_{\mathrm{m}}D\left(\frac{n_{\mathrm{f}}}{n_{\mathrm{m}}}\bigg|\frac{n_{\mathrm{r}}}{n_{\mathrm{m}}}\right). \qquad (13)$$

The connection distributions of the floppy and rigid phases obey the conditions that the relative entropy density from the rigid phase balances the density from the floppy one to the critical network and the entropic gain per unit volume of the floppy phase compensates the relative entropy from the rigid phase to the floppy one. Similarly, when any self-stress structure is forbidden, the phase boundary follows

$$n_{\mathrm{c}}\Lambda = n_{\mathrm{m}}D\left(\frac{n_{\mathrm{c}}}{n_{\mathrm{m}}}\bigg|\frac{n_{\mathrm{f}}}{n_{\mathrm{m}}}\right). \qquad (14)$$

The entropic gain per unit volume of the critical structure compensates the relative entropy from the floppy phase to the critical phase.

As derived and numerically verified in the Supplementary Note 5 and Supplementary Fig. 2, these balances, as well as the main results on the phase separation, hold in general for networks of multiple types of interactions, which is the case of real chalcogenides and proteins[40], as long as the vibrational entropy gain is approximately linear in probability distributions of interactions.

**Segregation in network glasses.** In network glasses, the degrees of freedom and the covalent constraints, both of which are associated with the atoms, depend differently on different chemical elements. The entropy-induced heterogeneous phase

develops by segregating different elements. For illustration purposes, we derive the phase boundaries of compounds $A_x B_{1-x}$, where $x$ is the number fraction of atoms $A$, the knob equivalent to the number of constraints $n$. Both $A$ and $B$ atoms, as isotropic particles, possess $d$ degrees of freedom. The number of constraints, counting both bond stretching and bond bending, satisfies $n^B < n_{\mathrm{c}} = d$ and $n^A > n_{\mathrm{c}}$, so that both floppy and rigid networks can be produced by different compositions. We perturb near the segregation of a stressed rigid phase of volume fraction $V_{\mathrm{r}}$, $B$ concentration $\rho_{\mathrm{r}}^B$ and $A$ concentration $\rho_{\mathrm{r}}^A$ and a floppy phase of $V_{\mathrm{f}}$, $\rho_{\mathrm{f}}^B$ and $\rho_{\mathrm{f}}^A$.

The vibrational entropy obeys,

$$\frac{S_{\mathrm{vib}}}{N} = V_{\mathrm{f}}\left[(n_{\mathrm{c}} - n^B)\rho_{\mathrm{f}}^B + (n_{\mathrm{c}} - n^A)\rho_{\mathrm{f}}^A\right]\Lambda, \qquad (15)$$

with $\Lambda$ the vibrational entropy gain from each floppy mode. The configurational entropy of two segregated regions is,

$$\frac{S_{\mathrm{conf}}}{N} = -V_{\mathrm{f}}\left(\rho_{\mathrm{f}}^B\ln\rho_{\mathrm{f}}^B + \rho_{\mathrm{f}}^A\ln\rho_{\mathrm{f}}^A\right) - V_{\mathrm{r}}\left(\rho_{\mathrm{r}}^B\ln\rho_{\mathrm{r}}^B + \rho_{\mathrm{r}}^A\ln\rho_{\mathrm{r}}^A\right). \qquad (16)$$

Optimizing entropy with the following constraints, $V_{\mathrm{f}} + V_{\mathrm{r}} = 1$, $V_{\mathrm{f}}\rho_{\mathrm{f}}^A + V_{\mathrm{r}}\rho_{\mathrm{r}}^A = x$, and $V_{\mathrm{f}}\rho_{\mathrm{f}}^B + V_{\mathrm{r}}\rho_{\mathrm{r}}^B = 1 - x$, we end up with following phase boundaries,

$$\rho_{\mathrm{f}}^A = \frac{e^{(n_{\mathrm{c}} - n^B)\Lambda} - 1}{e^{(n^A - n^B)\Lambda} - 1}; \qquad (17)$$

$$\rho_{\mathrm{r}}^A = \rho_{\mathrm{f}}^A e^{(n^A - n_{\mathrm{c}})\Lambda} = \frac{e^{(n^A - n^B)\Lambda} - e^{(n^A - n_{\mathrm{c}})\Lambda}}{e^{(n^A - n^B)\Lambda} - 1}; \qquad (18)$$

$$\rho_{\mathrm{f}}^B = 1 - \rho_{\mathrm{f}}^A; \quad \rho_{\mathrm{r}}^B = \rho_{\mathrm{f}}^B e^{(n^B - n_{\mathrm{c}})\Lambda} = 1 - \rho_{\mathrm{r}}^A; \qquad (19)$$

$$V_{\mathrm{r}} = \frac{x - \rho_{\mathrm{f}}^A}{\rho_{\mathrm{r}}^A - \rho_{\mathrm{f}}^A}. \qquad (20)$$

The boundary of the heterogeneous phase when self-stress is prohibited is determined by,

$$\Lambda' = \frac{1}{d}\left(\rho_{\mathrm{c}}^A\ln\frac{\rho_{\mathrm{c}}^A}{\rho_{\mathrm{f}}^A} + (1 - \rho_{\mathrm{c}}^A)\ln\frac{1 - \rho_{\mathrm{c}}^A}{1 - \rho_{\mathrm{f}}^A}\right) = \frac{1}{n_{\mathrm{c}}}D(\rho_{\mathrm{c}}|\rho_{\mathrm{f}}). \qquad (21)$$

As many constraints are associated with a high valence atom, the configurational entropy cost to generate phase separation is lower than in the network model by a factor of $1/n_{\mathrm{m}}$. So the transition boundary Eq. (21) is at a much lower value than Eq. (14), and the segregation occurs easier. In particular, we plot the phase diagram in Fig. 4c for chalcogenides $Ge_x Se_{1-x}$, where valences $r^{Se} = 2$ and $r^{Ge} = 4$ correspond to the number of covalent constraints $n^{Se} = 2$ and $n^{Ge} = 7$ counting both bond-stretching and bond-bending contributions[13]. Segregations occur above the critical point ($\Lambda_{\mathrm{c}} = 0$ $x_{\mathrm{c}} = 0.2$), and five phases with four homogeneous-heterogeneous transitions appear at the glass transition in varying $x$.

## Discussion

This comprehensive structural behavior provides a natural interpretation for the four transitions with discontinuous features, including transitions to the intermediate phase, as observed in chalcogenides when changing the chemical compositions[16]. Out of the intermediate phase, the micron-sized stress bubbles[15] are direct evidence of the heterogeneity. Its consequence on elasticity, the weakened shear modulus, is faithfully recorded in Raman

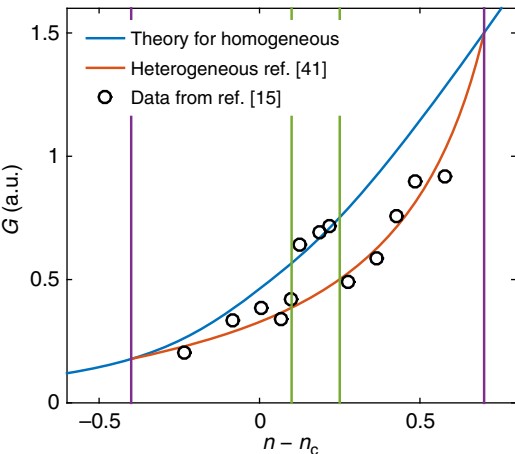

**Fig. 5** Experimental evidence of structural heterogeneity. Shear modulus $G$ in arbitrary unit predicted for homogeneous networks (blue) and for heterogenous networks composed of two phases at ends (red). Black circles are Raman shift data of the transverse optical (TO) branch $\nu_{TO}^2/\nu_0^2 - 1$ in ref.[15]. The green and purple lines indicate the boundaries of intermediate phase and separation range, as in Fig. 4

scattering experiments[15]. Distortions of micro-structures shift the Raman peaks proportional to the global elasticity, $\Delta \nu^2 \propto G$. As shown in Fig. 5, the jump of the Raman shift of the transversal optical branch in the intermediate phase[15] maps to the change of shear moduli between a homogeneous media and a heterogeneous mixture of two components[41]. In addition, high dynamical fragility out of the intermediate phase[16] is consistent with the appearance of very floppy structures[4], and the Einstein relation breaks down with a floppy-phase-dominated diffusion and a stressed-phase-limited relaxation[19], which results in a very stretched exponential relaxation[42].

According to the model, ruling the transitions is predominantly the entropic gain $\lambda$, which is negatively correlated with $\alpha$, the strength of the perturbing interactions relative to that of the strong ones forming the network. The width of the heterogeneous range is $\Delta n \propto \lambda \sim -\frac{1}{2} \ln \alpha$, whereas that of the homogeneous intermediate phase is $\Delta n \sim \sqrt{\alpha}$. Thus the larger is the entropic gain, that is, in terms of experimental parameters, the stronger are the covalent bonds or the weaker are the van der Waals forces, the easier is the glass being frozen in a heterogeneous structure and the narrower is the intermediate phase. This rule provides a general reference to the component-dependent widths of the intermediate phase[19]. Stabilizing the floppy parts as the weak interactions[43], the pressure should be another experimentally approachable knob. Starting from a heterogeneous structure, increasing pressure effectively increases $\alpha$ and leads to a transition to the homogeneous phase[18]. However, further pressure that distorts the strong interactions, $\alpha \sim 1$, breaks our premise on the separation of energy scales and thus ends up in new physics[19].

In conclusion, we have shown that the entropy favors heterogeneous structures in the vicinity of the rigidity threshold of networks. Based on the counting approximation[8,36,44], we have derived a phase diagram for the network model and found that the critical point rules the phase separation. A homogeneous intermediate phase emerges inside the heterogeneous separation range when stress energy becomes dominant at low temperature. The resulting transitions among heterogeneous and homogeneous phases potentially resolve the discontinuous features of the intermediate phase in chalcogenides[14–16]. The counting approximation simplifies the entropic gain as a single parameter

independent of the configurations. To go further, it is necessary to treat the entropic gain more carefully and study the global minimum and the dynamics toward it in a rougher free energy landscape induced by the complex entropic consequences of structures such as long chains. Meanwhile, it is important to test the separation in molecular dynamics simulations[17] for various temperatures and non-specific weak forces. Finally, it is useful to apply the role of entropy in protein foldings and self-assembly, where flexible units appear vital for elastic functions[5,45].

## Methods

**Metropolis algorithm and chosen parameters**. We equilibrate network structures $\Gamma$ using the Metropolis algorithm. From an initial configuration $\Gamma$, a new configuration is proposed by the random relocation of a spring, as illustrated in Fig. 2. By comparing the free energy Eq. (1) between the current and the new configurations, we sample and reset to the new configuration with probability $\min\left[1, \exp\left(-\frac{F(\Gamma')-F(\Gamma)}{T}\right)\right]$, where parameter $T$ defines the equilibrated temperature. For each combination of parameters $\{n, T, \alpha\}$, we implement in parallel 50 Monte Carlo simulations with $10^5$ steps to approach thermal equilibrium. When stress energy $H_0$ vanishes, $T$ is relevant only when thermal vibrations are so strong that Eq. (4) breaks down and nonlinear terms become important, discussed in Supplementary Discussion. In the model, we focus on the limit of the weak interactions $\alpha = 0.0003$[25,34]. In the segregation of chalcogenides, we apply $\alpha = 0.03$, a choice closer to the actual strength of van der Waals forces[4]. For the networks shown in Fig. 1d, from left to right, they are equilibrated at $n = 1.625$, $T = \alpha = 0.0003$; $n = 2.0$, $T = \alpha = 0.0003$; and $n = 2.25$, $T = 0.1$. To illustrate the floppy-isostatic separation in the model, we amplify the free energy loss by six times, an artifact unnecessary for segregation in chalcogenides.

**Data availability**. The authors declare that the data supporting the findings of this study are available within the article and its Supplementary Files, or are available from the authors upon reasonable request. The numerical data of the network model were generated by a home-written code on MATLAB interface. This code is available upon request.

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

## Acknowledgements

L.Y. thanks C. Jian, J. Liu, X. Mao, B. Shraiman and M. Wyart for discussions. L.Y. is supported by the Gordon and Betty Moore Foundation under Grant No. GBMF2919. This research was supported in part by the National Science Foundation under Grant No. NSF PHY-1748958. L.Y. also acknowledges the computation support from the "Center for Scientific Computing at UCSB" and NSF Grant CNS-0960316.

## Additional information

**Competing interests:** The authors declare no competing interests.

