## [Peer Review File · Nature Communications]

Reviewer #1 (Remarks to the Author):

In this manuscript, the author investigates the role of the entropy on the structural heterogeneity of models of amorphous materials exhibiting different topological/rigidity states. The author shows that heterogeneous structures are favored near the isostatic threshold.

Overall, the manuscript contains several interesting ideas that would deserve to be published in NatComm. However, the writing style sometimes makes it difficult to follow the thoughts of the author, e.g.:

"In fact, what is odd is rather the heterogeneous nature away from the threshold"

"It would be curious to see if changing pressure leads to similar heterogeneity-homogeneity transition"

I think the manuscript would be greatly improved by being further proofread.

My only concern with the manuscript lies in the attempt of applying the results to a Ge–Se glass. Although the "toy model" used by the author to establish his conclusion is very rich, I don't think it can meaningfully represent a "real glass." Indeed, one of the major assumptions of the model is that there is only one type of constraints, with a fixed stiffness. However, chalcogenide glasses always exhibit both radial bond-stretching (BS) and angular bond-bending (BB) constraints. BB constraints are expected to have a lower stiffness than that of BS constraints. How would such distinction affect the results?

Some minor points:

- How is the glass transition T_g of the system defined here?
- The numbers 4 and 14 of covalent constraints per atom for Se and Ge atoms are confusing as some of the constraints are shared between atoms. Based on their connectivity, one would expect Se and Ge atoms to have 2 (1 BS and 1 BB) and 7 (2 BS and 5 BB) constraints per atom, respectively.
- The propensity for amorphous structures to favor structural heterogeneity might explain the apparently contradictory observations regarding the existence of the intermediate phase. It would be great to comment on this.
- Can this model be used to explain the origin of the width of the intermediate phase (IP), that is, the extent of the compositional window over which self-organization is possible? For instance, why is the IP typically observed to be much wider in ternary chalcogenides than in binary glasses?

Reviewer #3 (Remarks to the Author):

This is a very interesting manuscript exploring the role of entropy at the liquid to disordered solid phase transition. Using both Monte Carlo simulations and analytic theory, the author investigated how entropy gain from floppy modes leads to a phase separation between floppy and rigid regions near the rigidity transition. The manuscript further discusses the competition between entropy and energy from self stress which explains the intermediate phase, as well as how the theory applies to chalcogenides. This manuscript presents original, convincing, and timely study on the nature of the glass transition problem, especially on the important role of vibrational entropy, which was often ignored in many studies. The manuscript is suitable for the broad audience of Nature Communications. I would recommend publication if the following questions are addressed:

1. I suppose that in the first part of the theory (before Eq.7) only entropy was considered for the phase separation and self stress energy was not included, but it's not clearly stated in the manuscript.
2. The paragraph before Eq.7 says "at finite T" but it seems to be actually adding in self stress energy compared to the part before it. The whole discussion before this part is also "at finite T". Also, is the last term in Eq.7 from self stress energy? How is it derived?
3. What are z_m , z_r , z_f and how is Eq.4 derived?
4. It was shown in the SI that $\Lambda > 0$. Why there is negative Λ part in the phase diagram, which also display a first order transition line?
5. The analytic theory presented in the paper made a number of approximations. For example, modes "belong" to floppy and rigid regions and no long range effects were considered. The contribution from self stress energy were also included in a simple way. Consequences of these approximations should be clearly discussed.
6. More broadly, the paper discusses the phase separation between floppy and rigid regions. What is the consequence of macroscopic properties? Fig 3D discusses elastic moduli and it can be seen from the simulation result that the shear modulus rises before $z=2d$, which probably originate from the percolation of the rigid region. What is the thermodynamic consequence of this at the glass transition?
7. The parameter Λ in the theory denotes the entropy gain of a floppy mode. What quantities does it relate to in an experimental system?

RESPONSE TO REVIEWERS:

I thank both reviewers for their detailed comments and constructive supports. In responding to concerns from the reviewers, all science-related changes are now colored in red.

Sincerely,
Le Yan

Reviewers' comments:

Reviewer #1 (Remarks to the Author):

In this manuscript, the author investigates the role of the entropy on the structural heterogeneity of models of amorphous materials exhibiting different topological/rigidity states. The author shows that heterogeneous structures are favored near the isostatic threshold.

Overall, the manuscript contains several interesting ideas that would deserve to be published in NatComm. However, the writing style sometimes makes it difficult to follow the thoughts of the author, e.g.:

"In fact, what is odd is rather the heterogeneous nature away from the threshold"

"It would be curious to see if changing pressure leads to similar heterogeneity-homogeneity transition"

I think the manuscript would be greatly improved by being further proofread.

I would like to first thank the reviewer for his/her interests and supports. On the writing, I have followed the suggestions from the Editor and used one of the suggested English language editing services to improve the manuscript. I hope it flows better now.

My only concern with the manuscript lies in the attempt of applying the results to a Ge-Se glass. Although the "toy model" used by the author to establish his conclusion is very rich, I don't think it can meaningfully represent a "real glass." Indeed, one of the major assumptions of the model is that there is only one type of constraints, with a fixed stiffness. However, chalcogenide glasses always exhibit both radial bond-stretching (BS) and angular bond-bending (BB) constraints. BB constraints are expected to have a lower stiffness than that of BS constraints. How would such distinction affect the results?

I appreciate the reviewer for this constructive concern. In the "toy model", the main assumption is the separation of energy scales, so that at the relevant temperature, we can start from the network of the strong interactions and consider the perturbation of the weak interactions, especially when the strong network becomes floppy. This point is now clarified when I introduce the network of springs of identical stiffness. As long as the separation of energy scales holds, our main results stand.

In the chalcogenides case, though the bond-bending (BB) constraints of covalent bonds appear weaker than the bond-stretching (BS) constraints, they are not as weak as van der Waals forces. So our assumption holds. To verify this special situation, I have performed the simulation with two types of strong springs $k_{BS}=k$ and $k_{BB}=0.5k$, now included in the Supplementary Information (SI). We find no quantitative difference in probability in the percolating cluster P_{∞} and observe the same phase separation.

In the discussion section, I have also pushed the theory to a more general case, when the stiffnesses of the strong springs obey a distribution, including in SI the simulation results of two special cases: a uniform distribution and a weakly diverging distribution (Gamma distribution with a shape parameter equals to 0.5). Based on the theory, I present a better physical understanding of the solutions Eqs.(5) (S28) and the entropic gain: the solutions are at the balance between the entropic gain and the relative entropy of the separated phases.

The only subtlety rising from the multiple types of stiffnesses is the potential separation of interactions. For the same reasoning deducing the separation of connectivity, the vibrational entropy may favor a separation of weaker interactions from the stronger ones, which is however not covered by the simplified theory. A detailed study of such separation is beyond the scope of the current manuscript and should be settled elsewhere.

Some minor points:

- How is the glass transition T_g of the system defined here?

I've been considering the glass transition temperature T_g in elastic models, reviewed in Ref. [39]. In these models, T_g is predicted to be proportional to the shear modulus. As one of the authors, I have shown numerically the proportional relation holds for the elastic network model (same as the one in this work) in Ref. [36], where T_g is defined by where the relaxation time of the Monte Carlo dynamics reaches a threshold. I now clarify in the text I'm using the convention of the elastic models for the glass transition temperature T_g , and derive the scaling relations used in the main text with a perturbation theory of the shear modulus in SI.

- The numbers 4 and 14 of covalent constraints per atom for Se and Ge atoms are confusing as some of the constraints are shared between atoms. Based on their connectivity, one would expect Se and Ge atoms to have 2 (1 BS and 1 BB) and 7 (2 BS and 5 BB) constraints per atom, respectively.

In the manuscript, I've consistently used z referring to the coordination number, which is different from the number of constraints per atom by a factor of two for two-body interactions. The coordination number indeed becomes confusing when multi-body interactions like bond-bending constraints appear. So now I have changed my notation to the number of constraints per atom throughout the manuscript.

- The propensity for amorphous structures to favor structural heterogeneity might explain the apparently contradictory observations regarding the existence of the intermediate phase. It would be great to comment on this.

The implications on the intermediate phase observed in chalcogenides are deeply discussed in the discussion section, including the macroscopic consequences of heterogeneity out of the intermediate phase and the origin of the width of the intermediate phase, as well as the possible effect of the pressure.

- Can this model be used to explain the origin of the width of the intermediate phase (IP), that is, the extent of the compositional window over which self-organization is possible? For instance, why is the IP typically observed to be much wider in ternary chalcogenides than in binary glasses?

The theory indicates that the width correlates positively with parameter alpha, the relative strength of the perturbing interactions that stabilize the structure. Such rule implies that the compounds of weaker covalent interactions or stronger van der Waals forces, usually associated with larger molecular weights, are likely to have wider intermediate phases. This is now in the discussion. However, I cannot conclude general laws for ternary or binary glasses from the theory. The tendency could be a result of stronger frustrations in ternary glasses that affect the glass transition.

Reviewer #3 (Remarks to the Author):

This is a very interesting manuscript exploring the role of entropy at the liquid to disordered solid phase transition. Using both Monte Carlo simulations and analytic theory, the author investigated how entropy gain from floppy modes leads to a phase separation between floppy and rigid regions near the rigidity transition. The manuscript further discusses the competition between entropy and energy from self stress which explains the intermediate phase, as well as how the theory applies to chalcogenides. This manuscript presents original, convincing, and timely study on the nature of the glass transition problem, especially on the important role of vibrational entropy, which was often ignored in many studies. The manuscript is suitable for the broad audience of Nature Communications. I would recommend publication if the following questions are addressed:

I first thank the reviewer for his/her strong supports.

1. I suppose that in the first part of the theory (before Eq.7) only entropy was considered for the phase separation and self stress energy was not included, but it's not clearly stated in the manuscript.

I have stated that the self-stress energy is not included when I introduce the corresponding numerical results just in front of the theory. I now emphasize this again at the beginning of the theory part by stating that: "When the self-stress energy is not participating, the balance between the vibrational entropy gain and the configurational cost determines the stability of the separation." When I recall the self-stress energy at the beginning of the section on intermediate phase, I add "when the energetic cost H_0

competes with the entropic gain, a homogeneous intermediate phase can develop inside the heterogeneous gap at low temperature.”

2. The paragraph before Eq.7 says "at finite T" but it seems to be actually adding in self stress energy compared to the part before it. The whole discussion before this part is also "at finite T". Also, is the last term in Eq.7 from self stress energy? How is it derived?

I have clarified at the beginning of the section that the self-stress energy is added. When the self-stress is not included, the temperature does not enter into the formalism. So here I was trying to recall the notation T of temperature, and clarify its unit. I now rephrase as “at temperature T” to avoid confusion. I have now included the derivation of the last term in Eq.(7) in the SI.

3. What are z_m , z_r , z_f and how is Eq.4 derived?

In responding to the suggestion from the first Reviewer, I have now changed all my conventions of the coordination number z to the number of constraints per atom n . n_m is the maximal available number of constraints, defined when I introduce the model on the triangular lattice, where $n_m=3$. n_r and n_f are the average numbers of constraints in the rigid and floppy phases correspondingly. They are introduced when we formalize the two-phase-separation layout and now illustrated in the Fig. 4a. The configuration entropy in Eq.(4) is just the entropy of mixing springs and vacancies summed over the floppy and rigid phases, I’ve now clarified in the text. There is also a two-phase boundary term that vanishes in the thermodynamic limit. More generally, when we have a continuous spectrum of spring types, the configuration entropy is the entropy of mixing all different types of springs and vacancies written in terms of the distribution of types, as added in the SI.

4. It was shown in the SI that $\Lambda > 0$. Why there is negative Λ part in the phase diagram, which also display a first order transition line?

I have now changed the way I introduce the parameter Λ to avoid confusion. In the formalism, parameter Λ is allowed to be any real number, while in the model, the entropy gain from each floppy mode (now as λ in lower case) is positive definite. However, when $\Lambda \leq 0$, the assumption of the model, the separation of scales, breaks down. So further work is required before extending the phase diagram to $\Lambda \leq 0$. Out of this consideration, I have now removed the negative Λ part in the phase diagram in Fig. 4a.

5. The analytic theory presented in the paper made a number of approximations. For example, modes "belong" to floppy and rigid regions and no long range effects were considered. The contribution from self stress energy were also included in a simple way. Consequences of these approximations should be clearly discussed.

The main approximations in the model include 1. the weak interactions as perturbations, 2. the counting approximation of contributions to the vibrational entropy and self-stress

energy of floppy and stressed modes (the approximation pointed out by the reviewer), 3. the mean-field approximation of the weak perturbations, and 4. the linear approximation of elasticity. 1. The perturbation approximation roots on our basic assumption on the separation of energy scales. Our model and derivations are based on the assumption. As now clarified in the manuscript, when this assumption breaks down, weak interactions become comparable to the strong ones, the perturbation approximation stops working and a new theory is necessary. 4. I have considered the situation in the SI Section H when the nonlinearity of constraints becomes relevant, the case at high temperature. On the contrary, the thermal volume is improved by long-range correlated motions and thus favors homogeneous structures. 3. The fact that the perturbations on different modes are not identical in fact has a similar indication as of the case when the counting approximation does not work, so I discuss them together.

2. I have shown that the counting approximation works in the vicinity of floppy-rigid separated phase. In this vicinity, the characteristic length of floppy modes and self-stress are considerably small, $l_c \sim \ln_{r-n_c} \{ \nu \}$, or $\ln_{f-n_c} \{ \nu \}$, as $n_r - n_c \sim 1$, and $n_c - n_f \sim 1$, so it's a good approximation to consider them localized and neglect the long-range correlation. The counting approximation stops working especially when the density of vibrations is sensitive to the configuration and contributes dominantly to the vibrational entropy. This is the case when the self-stress is prohibited. In such a situation, I have defined the entropic gain from the expansion near the optimal instead of counting modes. In general, the break down of the counting approximation does not change the vicinity of the maxima. So when the approximation breaks, the configuration-dependent entropic gain should lead to a rougher entropy landscape away from the maximal we are focusing here. In such case, finding the global minimum and the dynamics towards it are relevant topics to study. This has been added to the conclusive discussion.

6. More broadly, the paper discusses the phase separation between floppy and rigid regions. What is the consequence of macroscopic properties? Fig 3D discusses elastic moduli and it can be seen from the simulation result that the shear modulus rises before $z=2d$, which probably originate from the percolation of the rigid region. What is the thermodynamic consequence of this at the glass transition?

The consequences of heterogeneity associated with experimental features observed in chalcogenides are deeply discussed in the discussion section. The major consequences predicted are the weakened shear modulus compared to the homogeneous intermediate phase, heterogeneity-induced breakdown of Einstein relation, and stretched exponential relaxation. The discontinuity of the bulk modulus indeed originates from the percolation of the rigid region as discussed in the text. It indicates a rather complex consequence in glass: when the threshold n^* falls in the range of the intermediate phase, a weakening in the bulk modulus similar in shear modulus is expected at the heterogeneous transition, and when n^* is out of the intermediate phase, a strengthening of bulk modulus occurs instead. Otherwise, the existence of the weak forces smoothes any singular consequence of the rigidity percolation in real glasses.

The key point made in this work is that the singular features observed in experiments attribute to a structural separation transition rather than the rigidity transition.

7. The parameter Λ in the theory denotes the entropy gain of a floppy mode. What quantities does it relate to in an experimental system?

In the network model, Λ is linear in the logarithm of α , which is the relative strength of the weak non-specific interactions in the system. So both the strengths of the covalent bonds and van der Waals forces of the chemical components are relevant in an experimental system: the stronger the covalent bonds and the weaker the van der Waals forces, the wider range one can observe the features of the heterogeneous nature. In addition, the internal tension and pressure stabilize the network structure in a similar way to the weak interactions. So the entropic gain of a floppy mode will decrease when increasing the pressure in the experiment. Starting from the heterogeneous phase, a transition to homogeneous phase is expected. Increasing the pressure further to the energy density of the strong network, our description breaks down and further theoretical work is necessary. I have now included this discussion in the main text.

Reviewer #1 (Remarks to the Author):

The author has properly addressed all the comments, I recommend publication.

Reviewer #3 (Remarks to the Author):

The revised manuscript addressed all my questions, and the presentation is much improved. Thus I recommend publication.